# Serum Cartilage Oligomeric Matrix Protein and Golgi Protein-73: New Diagnostic and Predictive Tools for Liver Fibrosis and Hepatocellular Cancer?

**DOI:** 10.3390/cancers13143510

**Published:** 2021-07-13

**Authors:** Nikolaos K. Gatselis, Kalliopi Zachou, George Giannoulis, Stella Gabeta, Gary L. Norman, George N. Dalekos

**Affiliations:** 1Department of Medicine and Research Laboratory of Internal Medicine, National Expertise Center of Greece in Autoimmune Liver Diseases, General University Hospital of Larissa, 41110 Larissa, Greece; ngatsel@uth.gr (N.K.G.); zachouk@uth.gr (K.Z.); geogiannoulis7@gmail.com (G.G.); sgampeta@med.uth.gr (S.G.); 2Department of Research and Development, Inova Diagnostics, Inc., San Diego, CA 92131, USA; glnorman@inovadx.com

**Keywords:** biomarker, cartilage oligomeric matrix protein, golgi protein-73, aspartate aminotransferase/platelets ratio index score, Fibrosis-4 score, hepatic fibrosis, cirrhosis, hepatocellular carcinoma

## Abstract

**Simple Summary:**

Hepatocellular carcinoma (HCC) ranks as the sixth most common malignancy and represents the fourth leading cause of cancer-related deaths. However, most HCC cases are insidious in the early stages leading to a delay in diagnosis with limited treatment options. In patients with chronic liver diseases, advanced liver fibrosis and cirrhosis are the leading risk factors for the development of HCC. Cartilage oligomeric matrix protein (COMP) and Golgi protein-73 (GP73) are two biomarkers that have been associated with the progression of chronic liver disease, including inflammation, fibrosis, and HCC. The aim of our study was to assess the performance of the combination of these biomarkers. We confirmed, in a large cohort of 288 patients with chronic liver diseases, that the combination of GP73 and COMP had a high discriminative ability to detect severe fibrosis/cirrhosis and is efficient in predicting the development of HCC.

**Abstract:**

The cartilage oligomeric matrix protein (COMP) and Golgi-protein-73 (GP73) have been proposed as markers of liver fibrosis and hepatocellular carcinoma (HCC). Our aim was to assess the performance of the combination of these markers in diagnosing cirrhosis and predicting HCC development. Sera from 288 consecutive patients with chronic liver diseases were investigated by using COMP and GP73-ELISAs. Dual positivity for COMP (>15 U/L) and GP73 (>20 units) was observed in 24 (8.3%) patients, while 30 (10.4%) were GP73(+)/COMP(−), 37/288 (12.8%) GP73(−)/COMP(+), and 197 (68.5%) GP73(−)/COMP(−). Positivity for both markers was associated with cirrhosis [23/24 (95.8%) for GP73(+)/COMP(+) vs. 22/30 (73.3%) for GP73(+)/COMP(−) vs. 25/37 (67.6%) for GP73(−)/COMP(+) vs. 46/197 (23.4%) for GP73(−)/COMP(−); *P* < 0.001]. The combination of GP73, COMP, the aspartate aminotransferase/platelets ratio index, and the Fibrosis-4 score had even higher diagnostic accuracy to detect the presence of cirrhosis [AUC (95% CI): 0.916 (0.878–0.946)] or significant liver fibrosis (METAVIR ≥ F2) [AUC (95% CI): 0.832 (0.768–0.883)] than each marker alone. Kaplan-Meier analysis showed that positivity for both GP73 and COMP was associated with higher rates of HCC development (*P* < 0.001) and liver-related deaths (*P* < 0.001) during follow-up. In conclusion, the combination of GP73 and COMP seems efficient to detect cirrhosis and predict worse outcomes and the development of HCC in patients with chronic liver diseases.

## 1. Introduction

Hepatocellular carcinoma (HCC) ranks as the sixth most common malignancy and represents the fourth leading cause of cancer-related deaths worldwide [1]. HCC usually arises as a complication of end-stage liver disease, secondary to either chronic viral hepatitis or other non-viral chronic liver diseases [2]. In patients with chronic liver diseases, advanced liver fibrosis and cirrhosis are the leading risk factors for the development of HCC [3].

Liver fibrosis is the final result of chronic inflammation in the liver in most chronic liver diseases independent of etiology, leading to severe structural and functional changes of liver parenchyma. In addition, the fibrosis stage is one of the main prognostic factors of the outcome, as it predicts the development of cirrhosis and HCC [4,5,6,7]. However, most cases of HCC are insidious in the early stages resulting in the diagnosis of HCC in a significant proportion of patients at an advanced stage, which has generally poor prognosis with limited treatment options [8]. Therefore, the early recognition and effective treatment of chronic liver diseases along with surveillance strategies are crucial to improve the overall survival of patients [4,5].

At present, liver biopsy remains the best standard for accurate staging, also providing additional information such as the severity of necroinflammation, the presence of steatosis, or other concomitant factors, which may contribute to liver injury [9]. However, it carries potential limitations including the risk of complications, sampling error, and interobserver variations [10]. Therefore, the utility of routine liver biopsy has been reduced by the use of non-invasive serological or imagine methods [6], such as the AST-to-platelet ratio index (APRI) [11], Fibrosis-4 score (FIB-4) [12], and transient elastography (TE) [13,14]. Alpha-fetoprotein (AFP), the most frequently used serum biomarker in the surveillance and diagnosis of HCC in clinical practice, is characterized by low diagnostic performance, especially during early-stage HCC, indicating an unmet need for developing effective new biomarkers for HCC detection, which could complement or even replace AFP [4,5,15].

Recently, studies from our group and others have shown that serum cartilage oligomeric matrix protein (COMP) and Golgi protein-73 (GP73) levels were positively correlated with indices of progression of chronic liver diseases, such as inflammatory activity, fibrosis/cirrhosis, and HCC [16,17,18,19,20,21,22,23,24,25]. COMP is largely found in the extracellular matrix protein of skeletal tissue, but increased COMP expression has been also associated with fibrogenesis in patients with cirrhosis and HCC [17,23,24,25]. GP73 is a 73 kDa transmembrane glycoprotein mainly expressed in biliary epithelial cells but rarely in hepatocytes in the normal liver [26]. However, GP73 expression is considerably upregulated in hepatocytes in patients with either acute or chronic liver disease, including HCC [27,28].

We conducted this study in an attempt to evaluate the performance of the combination of these markers in detecting cirrhosis and predicting the development of HCC. For this reason, we investigated a large group of 288 consecutive patients with chronic hepatopathies followed in our department as similar data are missing.

## 2. Material and Methods

The study population included 129 patients with chronic hepatitis B (CHB), 120 with chronic hepatitis C (CHC), 12 with alcoholic liver disease, 21 with primary biliary cholangitis (PBC), and 6 with autoimmune hepatitis (AIH). Patients with a concurrent history of autoimmune rheumatic diseases or osteoarthritis at baseline and during follow-up were excluded from the analysis. All patients were prospectively diagnosed and followed in our center between 2000 and 2013. At initial evaluation, we collected serum samples from all patients (*n* = 288), which were stored at −20 °C until the investigations for COMP and GP73. Liver stiffness measurements (LSM) and liver biopsies at the same time point as the collection of serum samples were available in 177 and 127 patients, respectively. The baseline characteristics of patients are shown in Table 1.

At baseline, 116/288 patients (40.3%) were cirrhotic, 33/116 (28.4%) had already experienced at least one episode of decompensation in the past, and 12/288 patients (4.2%) were diagnosed with HCC at initial evaluation. Further, 236 out of 288 patients were followed ≥6 months for a median (interquartile range (IQR)) duration of 98 (92) months. The mean ± standard deviation (SD) follow-up period was 102 (±66) months. During follow-up, 10/148 (6.8%) patients developed cirrhosis and 23/98 (23.5%) cirrhotic patients developed the first episode of decompensation. Among 236 patients with long-term follow-up (≥6 months), 7 diagnosed with HCC at baseline and 32 developed HCC subsequently during follow-up (Figure 1). Finally, a liver-related death occurred in 53/288 (18.4%) in the total group of patients and 45/236 (19.1%) in the group of patients with follow-up ≥6 months. None of the patients received a liver transplant during the follow-up period of the study.

Cirrhosis was diagnosed according to our previous publications. In brief, in patients without liver biopsy, cirrhosis was diagnosed by liver ultrasound (coarse echo-pattern of the liver in association with irregular margins of the parenchyma, spleen length above 12 cm, portal vein diameter above 16 mm), TE, supporting findings in endoscopy (esophageal varices, portal gastropathy), and/or clinical signs of decompensated cirrhosis (presence of ascites, history of variceal bleeding, hepatic encephalopathy) [16,17,18,29,30]. The diagnosis of HCC was made by standard findings on histology and/or radiology [4,5]. One hundred and seventy-seven patients had an available liver biopsy and fibrosis stage according to the METAVIR score [31]. The fibroscan device powered by VCTE (ECHOSENS^®^, Paris, France) using the standard M probe was used for LSM determinations. The results were expressed as median (kPa) of 10 valid determinations with the associated IQR and the success rate. An index LSM was considered valid if the IQR was less than 0.3 [32]. The surveillance for HCC development was done by ultrasound of the upper abdomen and AFP determination every 6 months in patients with established cirrhosis and every 12 months in those without cirrhosis. During follow-up, the development of cirrhosis, decompensation of liver disease, and HCC development were assessed according to international criteria [4,5]. Furthermore, we recorded the cause of deaths.

The prototype GP73 ELISA (QUANTA Lite^®^ GP73, Inova Diagnostics, Inc., San Diego, CA, USA, Research Use Only) was used for the determination of GP73 as we described previously [16]. Briefly, this assay uses novel anti-GP73 monoclonal and rabbit polyclonal antibodies against the full-length GP73 antigen [16]. According to the manufacturer’s instructions, serum samples with more than 20 arbitrary units were considered positive as this cut-off has been established in preliminary experiments in 518 healthy and disease controls [16].

An ELISA technique (AnaMar Medical, Lund, Sweden) was used for serum COMP determinations [17,18]. As we have described previously, this assay uses two monoclonal mouse antibodies against two antigenic determinants of COMP [17,18]. Five calibrators corresponding to 0.4, 0.7, 1.2, 1.8, and 3.2 U/L were used for the calibration curve. We have demonstrated that the more rigorous and specific cut-off of 15 U/L is more appropriate for patients with chronic hepatopathies [17].

Alanine aminotransferase (ALT), aspartate aminotransferase (AST), alkaline phosphatase (ALP), bilirubin, γ-glutamyl transpeptidase (γ-GT), albumin, international normalized ratio (INR), platelets (PLT), and AFP were determined by standard techniques. APRI and FIB-4 scores were also determined at baseline [11,12].

### Statistical Analysis

The Kolmogorov-Smirnov test was used to assess the normality of the distribution of variables. Quantitative values are expressed as median (IQR). The Pearson’s Chi-Square, Kruskal-Wallis test, Spearman’s rho correlation, Mann-Whitney U test, multivariate logistic regression, and multivariate Cox regression analyses were used for the analysis of the data, where appropriate. For the combined effect of diagnostic biomarkers (COMP, GP73, APRI, and FIB-4), we used the logistic regression predicted value probability. Receiver operating characteristic (ROC) curves were constructed, and comparisons were done by the DeLong test. The Kaplan-Meier plots according to the detection of COMP and GP73, up to the end of follow-up or when patients reached one of the endpoints (development of cirrhosis, liver decompensation, HCC development, liver-related death) were used for survival. Two-sided *P*-values < 0.05 were considered statistically significant. All data analyses were done with SPSS version 26.0.

## 3. Results

### 3.1. Significance of Dual Positivity for GP73 and COMP in Relation to the Baseline Characteristics of Patients

Table 2 presents the baseline characteristics of patients according to COMP and GP73 positivity. GP73 levels were positively correlated with age (r = 0.380; *P* < 0.001), INR (r = 0.365; *P* < 0.001), AST (r = 0.417; *P* < 0.001), ALT (r = 0.235, *P* < 0.001), γ-GT (r = 0.309, *P* < 0.001), ALP (r = 0.224, *P* < 0.001), bilirubin (r = 0.367; *P* < 0.001), and AFP (r = 0.242; *P* < 0.001), while they were negatively associated with platelets (r = −0.429; *P* < 0.001) and albumin (r = −0.545; *P* < 0.001). Moreover, GP73 levels were higher in patients with fibrosis stage ≥ F2 [11.2 (9.8) vs. 7.7 (4.1); *P* < 0.001] or stage ≥ F3 [13.7 (12.6) vs. 8.1 (4.9); *P* < 0.001], as well as in cirrhotic [16.2 (15.6) vs. 8.3 (4.5); *P* < 0.001], decompensated [22.9 (15.9) vs. 14.7 (10.3); *P* < 0.001], and HCC patients [13.6 (18.5) vs. 10.3 (9); *P* = 0.017].

Similarly, a positive correlation was found between COMP levels and age (r = 0.421; *P* < 0.001), INR (r = 0.294; *P* < 0.001), AST (r = 0.394; *P* < 0.001), ALT (r = 0.225, *P* < 0.001), γ-GT (r = 0.226, *P* < 0.001), ALP (r = 0.286, *P* < 0.001), bilirubin (r = 0.181; *P* = 0.004), AFP (r = 0.339, *P* < 0.001), while a negative correlation was noticed with platelets (r = −0.294; *P* < 0.001) and albumin (r = −0.315, *P* < 0.001). Additionally, COMP levels were increased in patients with fibrosis stage ≥ F2 [11 (5) vs. 8.9 (4.7); *P* < 0.001] or fibrosis stage ≥ F3 [11.8 (6.5) vs. 9.3 (4.9); *P* < 0.001], as well as in cirrhotic [13.7 (7.9) vs. 9.3 (7.9); *P* < 0.001] and HCC patients [14 (16.3) vs. 10.2 (6); *P* = 0.028]. No difference was found between COMP levels, in regard to the presence of decompensation in cirrhotic patients.

Dual positivity for COMP (>15 U/L) and GP73 (>20 units) was detected in 24/288 (8.3%) patients, while 30/288 (10.4%) were GP73 positive only, 37/288 (12.8%) were COMP positive only, and 197/288 (68.4%) patients tested negative for both markers. Positivity for both markers was characterized by increased age (*P* < 0.001), higher INR (*P* < 0.001), AST (*P* < 0.001), ALT (*P* < 0.001), γ-GT (*P* < 0.001), ALP (*P* < 0.001), bilirubin (*P* < 0.001), AFP (*P* < 0.001), and lower platelets (*P* < 0.001) and albumin (*P* < 0.001) levels (Table 2).

Of note, the combined positivity for GP73 and COMP were associated with the presence of cirrhosis at baseline; 23/24 (95.8%) for GP73(+)/COMP(+) vs. 22/30 (73.3%) for GP73(+)/COMP(−) vs. 25/37 (67.6%) for GP73(−)/COMP(+) vs. 46/197 (23.4%) for GP73(−)/COMP(−) (P < 0.001). APRI score was higher in GP73(+)/COMP(+) patients (*P* < 0.001). In patients with available liver biopsy (*n* = 177), positivity for each or both markers was associated with METAVIR ≥ F2 stage of fibrosis; 8/8 (100%) for GP73(+)/COMP(+) vs. 14/15 (93.3%) for GP73(+)/COMP(−) vs. 16/17 (94.1%) for GP73(−)/COMP(+) vs. 76/137 (55.5%) for GP73(−)/COMP(−) (*P* < 0.001, Table 2).

Multivariate logistic regression analysis revealed that dual combination of COMP plus GP73 was the most potent independent predictive factor for the presence of cirrhosis (odds (OR) = 53, 95% CI: 11.99–233.9; *P* < 0.001) even after adjustment for age (OR = 1.053, 95% CI: 1.022–1.085; *P* = 0.001), male sex (OR = 3.114, 95% CI: 1.388–6.987; *P* = 0.006), AST/ALT levels (OR = 2.076, 95% CI: 0.890–4.839; *P* = 0.091), platelets (OR = 0.016, 95% CI: 0.002–0.147; *P* < 0.001), γ-GT (OR = 1.007, 95% CI: 1.000–1.014; *P* = 0.036), and categorization of liver disease as viral or non-viral (OR = 0.751, 95% CI: 0.216–2.611).

At baseline, 12 patients were diagnosed with HCC. The diagnostic performance of AFP to diagnose HCC showed 67% sensitivity, 89% specificity, 21% positive predictive value, and 98% negative predictive value. The triple combined serum positivity for COMP, GP73, and AFP showed 17% sensitivity, 98% specificity, 25% positive predictive value, and 96% negative predictive value.

### 3.2. Dual Positivity for GP73 and COMP as a Diagnostic Marker of Significant Fibrosis and Cirrhosis

Dual positivity for GP73 and COMP had a high discriminative ability for detecting cirrhosis [AUC (95% CI): 0.882 (0.839–0.917)]. Furthermore, the combination of GP73, COMP, APRI, and FIB-4 had the highest diagnostic accuracy to detect the presence of cirrhosis [AUC (95% CI): 0.916 (0.878–0.946)] than each marker alone: GP73 [AUC (95% CI): 0.843 (0.795–0.883); *P* < 0.001], COMP [AUC (95% CI): 0.770 (0.716–0.817); *P* < 0.001], APRI [AUC (95% CI): 0.833 (0.784–0.874); *P* < 0.001], and FIB-4 [AUC (95% CI): 0.885 (0.846–0.920); *P* = 0.02] (Figure 2). Likewise, the triple combination of COMP, GP73, and FIB-4 [AUC (95% CI): 0.913 (0.874–0.943)] was superior to each marker alone: COMP (*P* < 0.001), GP73 (*P* = 0.001), APRI (*P* < 0.001), and FIB-4 (*P* = 0.025).

Similarly, in the subgroup of patients with available liver biopsy (*n* = 177), positivity for both GP73 and COMP had a high discriminative ability for detecting significant (≥F2) fibrosis [AUC (95% CI): 0.789 (0.722–0.847)]. The combination of GP73, COMP, APRI, and FIB-4 improved the diagnostic performance further [AUC (95% CI): 0.832 (0.768–0.883)] compared to each marker alone: GP73 [AUC (95% CI): 0.742 (0.671–0.804); *P* = 0.01], COMP [AUC (95% CI): 0.708 (0.635–0.774); *P* < 0.001], APRI [AUC (95% CI): 0.762 (0.692–0.822); *P* = 0.068], and FIB-4 [AUC (95% CI): 0.799 (0.732–0.865); *P* = 0.171] (Figure 3). The triple combination of COMP, GP73, and FIB-4 [AUC (95% CI): 0.829 (0.765–0.881)] also performed better than each marker alone: COMP (*P* = 0.039), GP73 (*P* = 0.037), APRI (*P* = 0.036), and FIB-4 (*P* = 0.034). In addition, the quadrable combination of GP73, COMP, APRI, and FIB-4 retained the highest diagnostic accuracy to detect ≥F3 fibrosis [AUC (95% CI): 0.844 (0.783–0.894)] (Appendix A).

Finally, in patients with available LSM by TE (*n* = 127), the combined GP73 and COMP positivity had a high discriminative ability to detect significant (≥F2) fibrosis [AUC (95% CI): 0.791 (0.710–0.858)]. The triple combination of GP73, COMP, and FIB-4 improved the diagnostic performance further [AUC (95% CI): 0.841 (0.766–0.900)] compared to each marker alone: GP73 [AUC (95% CI): 0.676 (0.587–0.756); *P* < 0.001], COMP [AUC (95% CI): 0.772 (0.690–0.842); *P* = 0.100], APRI [AUC (95% CI): 0.759 (0.675–0.830); *P* = 0.052], and FIB-4 [AUC (95% CI): 0.802 (0.722–0.868); *P* = 0.210] (Figure 4). In addition, the GP73, COMP, and FIB-4 combination retained the highest ability to detect fibrosis stage ≥F3 (Appendix A).

The comparison of patients with increased AFP (>10 ng/mL) to those with normal levels (≤10 ng/mL) did not show any significant difference in the diagnostic accuracy of the GP73 and COMP combination for the detection of cirrhosis [AUC (95% CI): 0.835 (0.688–0.981) vs. 0.864 (0.812–0.917); *P* = 0.713], severe fibrosis (≥F3) based on liver biopsy [AUC (95% CI): 0.854 (0.661–1.047) vs. 0.760 (0.678–0.843); *P* = 0.382], or severe fibrosis (≥F3) based on TE measurements [AUC (95% CI): 0.778 (0.490–1.066) vs. 0.801 (0.721–0.881); *P* = 0.880].

The diagnostic performance of various combinations between COMP, GP73, APRI, and FIB-4 score for detecting cirrhosis or significant fibrosis is shown in Appendix A.

### 3.3. Combination of GP73 and COMP as Predictive Marker of Outcome

The development of cirrhosis was not different in GP73(+)/COMP(+), GP73(+)/COMP(−), GP73(−)/COMP(+), and GP73(−)/COMP(−) patients (Kaplan-Meier analysis; *P* = 0.589). Similarly, the development of liver decompensation during follow-up in patients with compensated cirrhosis at baseline and without any previous episode of decompensation was not different in the four patient groups (*P* = 0.313).

In contrast, patients with positivity for both GP73 and COMP who were followed for more than 6 months had significantly higher rates of HCC development either in the total patients’ group (*n* = 229; *P* < 0.001; Figure 5A) or in those with established cirrhosis at baseline (*n* = 82; P < 0.001). Sub-analysis, comparing only GP73(+)/COMP(+) vs. GP73(+)/COMP(−) and GP73(+)/COMP(+) vs. GP73(−)/COMP(+), further verified the higher performance of dual positivity in predicting development of HCC during follow-up (Kaplan-Meier analysis; *P* = 0.005 and *P* = 0.001, respectively). Multivariate Cox regression analysis showed that the combination of COMP and GP73 was by far the strongest negative predictive factor for the development of HCC [hazard ratio (HR) = 13.55, 95% CI: 3.616–50.78; *P* < 0.001] even after adjustment for other known risk factors for HCC development such as age, male sex, platelets, and FIB-4 (Table 3). The significance of the COMP and GP73 combination as a predictor for HCC development also remained significant [HR = 29.4, 95% CI: 7.202–120.3; *P* < 0.001] in patients with normal AFP (≤10 ng/mL).

Of note, among 229 non-HCC patients at baseline who were followed, the AFP positivity showed a 41% sensitivity, 91% specificity, 42% positive predictive value, and 90% negative predictive value for HCC development during follow-up while the respective indices of triple seropositivity for COMP, GP73, and AFP were 13%, 99%, 67%, and 87%.

To investigate if an increase of COMP or GP73 levels further increases the likelihood of HCC development, we stratified each biomarker into three categories: (i) GP73 or COMP ≤ mean + SD, (ii) mean + SD < GP73 or COMP ≤ mean + 2SD, and (iii) GP73 or COMP > mean + 2SD. By using GP73 ≤ mean + SD as the reference group, we found that an increase of GP73 levels raises the likelihood for HCC development: HR = 7.703, 95% CI: 2.849–20.82 for the “GP73 > mean + SD” group (*P* < 0.001) and HR = 10.8, 95% CI: 3.699–31.543 for the “GP73 > mean + 2SD” group (*P* < 0.001). Regarding COMP, a further increase of likelihood for HCC development was noticed only for the group of “COMP > mean + SD” [HR = 4.686, 95% CI: 2.012–10.91 (*P* < 0.001)]. In contrast, the GP73/COMP ratio was not associated with the development of HCC (HR = 1.075, 95% CI: 0.942–1.228, *P* = 0.283).

Furthermore, patients with positivity for both GP73 and COMP and long-term follow-up (≥6 months) were characterized by significantly higher rates of liver-related deaths (*P* < 0.001; Figure 5B). Multivariate Cox-regression analysis revealed the combination of COMP with GP73 as the most potent negative predictive factor of liver-related mortality [HR = 23.11, 95% CI: 6.972–76.58; *P* < 0.001] even after an adjustment for confounding factors, which have been revealed from the univariate analysis (Table 4).

## 4. Discussion

Our study showed that dual positivity for GP73 and COMP is strongly associated with the presence of significant fibrosis and cirrhosis. Importantly, all but one patient (23/24, 95.8%) were positive for both markers were cirrhotic. In addition, this combination appears to be the strongest independent predictive marker for the development of HCC (HR = 13.55) and the most potent independent predictive factor of liver-related mortality (HR = 23.11). Furthermore, our analysis demonstrated that the combination of GP73, COMP, and APRI or GP73, COMP, and FIB-4 yielded the highest diagnostic accuracy to detect significant (≥F2) fibrosis and cirrhosis, compared to each marker alone.

The assessment of liver fibrosis in patients with chronic liver diseases is fundamental, as it is associated with the risk of cirrhosis and liver-related complications including HCC development [33]. Easily determined serum biomarkers have gathered attention as potentially attractive diagnostic tools for the diagnosis of cirrhosis because they are based on simple blood tests. In addition, their interpretation seems reliable taking into account that cut-offs with high sensitivity and specificity can be defined [6]. For example, the APRI score is well-validated for the assessment of the fibrosis stage, based on the determination of simple markers, such as platelets and AST. However, its diagnostic accuracy seems superior in diagnosing cirrhosis than the detection of lower fibrosis stages [34,35]. For these reasons, early stages of fibrosis and the identification of patients at risk for progression to advanced liver disease cannot be identified with certainty, resulting in a significant delay in the implementation of reliable surveillance and rigorous management [6]. On the other hand, previous reports from our group have shown that GP73 and COMP were strongly associated with liver cirrhosis and can predict the outcome of patients with chronic liver diseases as attested by the identification of patients who were at increased risk for HCC development and liver-related death [16,17,18,19].

GP73 is upregulated in many chronic hepatopathies, including chronic viral hepatitis B and C, alcohol-related liver disease, and AIH [36,37,38,39]. Additionally, GP73 levels were considerably elevated in patients with significant necroinflammation, fibrosis, or cirrhosis [36,37,40,41,42], suggesting that GP73 could be used as a reliable surrogate marker for the detection of advanced fibrosis and cirrhosis but also for the monitoring of patients with liver diseases [16,37,40,43,44,45]. Indeed, recent reports have shown that serum GP73 is an accurate biomarker for the detection of significant fibrosis, bearing higher diagnostic accuracy than APRI and FIB-4 scores and that it could be used as a complementary tool to TE [20,46].

The investigation of COMP in the sera of patients was performed first in patients with rheumatoid arthritis and osteoarthritis in an attempt to evaluate the destruction of the cartilage [47,48,49]. This initial concept raised concerns about its specificity in detecting cirrhosis and the early development of HCC. However, subsequent studies by Xiao et al. [45] demonstrated that COMP is slightly expressed in cirrhotic liver and overexpressed in HCC, while recently, Magdaleno et al. [23] showed that COMP contributes to the progression of liver fibrosis by regulating collagen deposition. Interestingly in the latter study, the authors were able to demonstrate how the absence of COMP was associated with decreased hepatic inflammation and fibrosis in two animal models of induced liver fibrosis. As patients with chronic liver diseases and concurrent autoimmune rheumatic diseases or osteoarthritis have been excluded from our current and previous study [17], our findings suggest that COMP fragments could be detected in the serum samples of patients with liver diseases during the fibrogenic process, potentially representing the level of the fibrogenic activity in the liver. Furthermore, it has been reported that COMP is highly expressed in HCC tumor cells, indicating that it might also play a pathophysiological role in hepatocarcinogenesis [25].

In general, the combination of various biomarkers in order to further improve the diagnostic accuracy for detecting severe fibrosis and cirrhosis as well as to facilitate the early detection of HCC seems promising and attractive [50,51,52]. Several combinations have been proposed, such as the SAFE biopsy algorithm for the non-invasive assessment of liver fibrosis, which consists of APRI and a commercialized method (Fibrotest-Fibrosure) [53] or the combination of GP73, glypican-3, and AFP for the diagnosis of HCC [54]. However, several aspects should be taken into account when using these combination markers, such as the cut-off levels, characteristics of the control groups used during their validation, the tumor burden of patients, the exact algorithm, and whether the same algorithm has been applied in other studies. Another important point is that while the combinations result in higher specificity, a decrease in sensitivity will inevitably occur [52].

Our study provides important evidence on this issue, suggesting that the combination of GP73 and COMP could be used to assess the presence of cirrhosis in patients with chronic liver diseases simply and inexpensively by ELISAs and most importantly, contrary to other more expensive and complex models for fibrosis staging [55]. Notably, most patients with dual positivity for COMP and GP73 had cirrhosis, while in the subgroup analysis of patients with available liver biopsy, all patients who tested positive for both biomarkers had significant fibrosis. In addition, the combined reactivity against COMP and GP73 antigens seems to be associated with a worse outcome of patients as attested by the detection of patients at high risk for progression to HCC development and higher liver-related mortality. Although the type of antiviral treatment in patients with viral hepatitis might eventually affect the rates of HCC development and decompensation of liver disease, the very long follow-up period up to the first trimester of 2021, with a median (IQR) duration longer than 8 years, seems to ameliorate the impact of these different treatment strategies for patients with viral hepatitis.

## 5. Conclusions

Our study showed quite convincingly that the combined determination of GP73 and COMP by simple ELISAs had a high discriminative ability in order to detect patients with severe fibrosis/cirrhosis, while it was also efficient to independently predict the worse outcome of patients and the development of HCC. These important and novel findings need to be confirmed in larger multicenter studies.

## Figures and Tables

**Figure 1 cancers-13-03510-f001:**
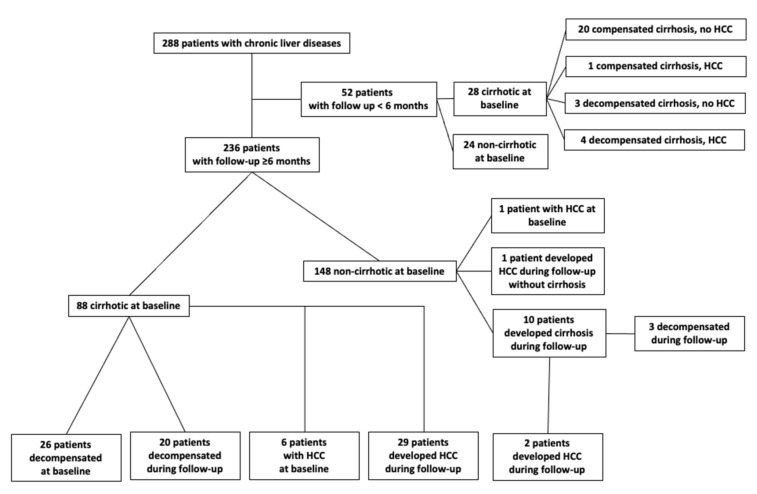
Flow-chart of the 288 patients that participated in the study and liver-related events observed during follow-up.

**Figure 2 cancers-13-03510-f002:**
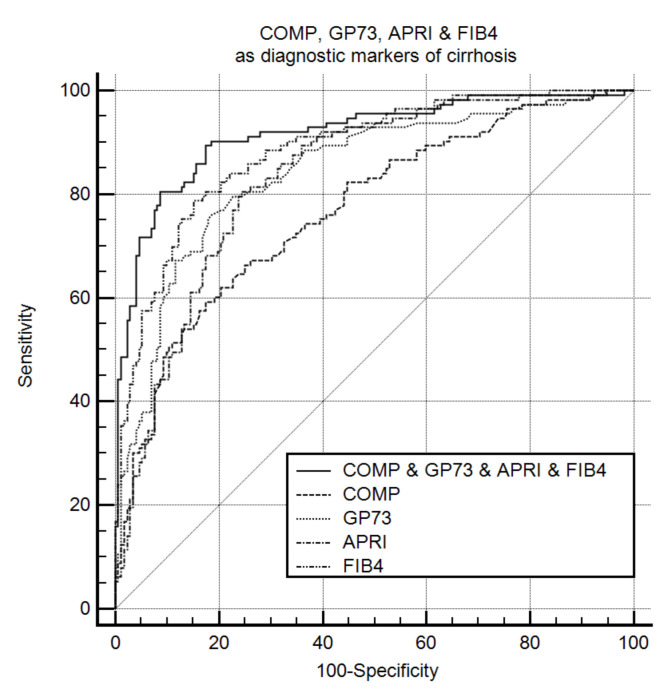
COMP, GP73, APRI, and FIB-4 as diagnostic markers of cirrhosis (*n* = 288). Receiver operating characteristic curves for the prediction of cirrhosis according to COMP, GP73, APRI, and FIB-4. The combination of COMP, GP73, APRI, and FIB-4 had the highest diagnostic accuracy for cirrhosis [AUC (95% CI): 0.916 (0.878–0.946)] than each marker alone: GP73 [AUC (95% CI): 0.843 (0.795–0.883); *P* < 0.001], COMP [AUC (95% CI): 0.770 (0.716–0.817); *P* < 0.001], APRI [AUC (95% CI): 0.833 (0.784–0.874); *P* < 0.001], and FIB-4 [AUC (95% CI): 0.885 (0.846–0.920); *P* = 0.02]. COMP: Cartilage oligomeric matrix protein. GP73: Golgi protein 73. APRI: Aspartate aminotransferase to platelets index. FIB-4: Fibrosis-4. AUC: Area under the curve.

**Figure 3 cancers-13-03510-f003:**
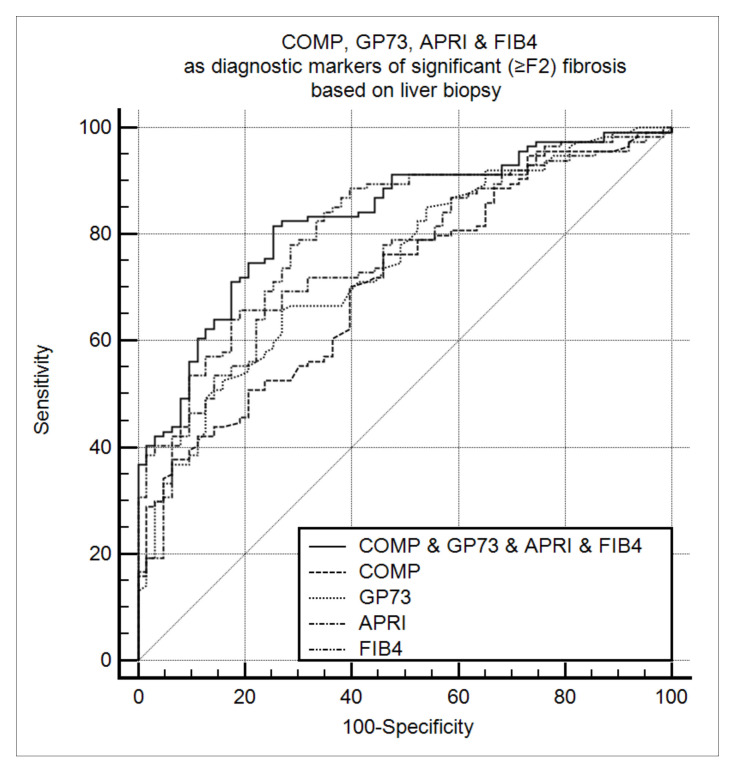
COMP, GP73, APRI, and FIB-4 as diagnostic markers of significant fibrosis (≥F2) according to liver biopsy (*n* = 177). The combination of COMP, GP73, APRI, and FIB-4 had the highest diagnostic performance [AUC (95% CI): 0.832 (0.768–0.883)] compared to each marker alone: GP73 [AUC (95% CI): 0.742 (0.671–0.804); *P* = 0.01], COMP [AUC (95% CI): 0.708 (0.635–0.774); *P* < 0.001], APRI [AUC (95% CI): 0.762 (0.692–0.822); *P* = 0.068], and FIB-4 [AUC (95% CI): 0.799 (0.732–0.865); *P* = 0.171]. COMP: Cartilage oligomeric matrix protein. GP73: Golgi protein 73. APRI: Aspartate aminotransferase to platelets index. FIB-4: Fibrosis-4. AUC: Area under the curve.

**Figure 4 cancers-13-03510-f004:**
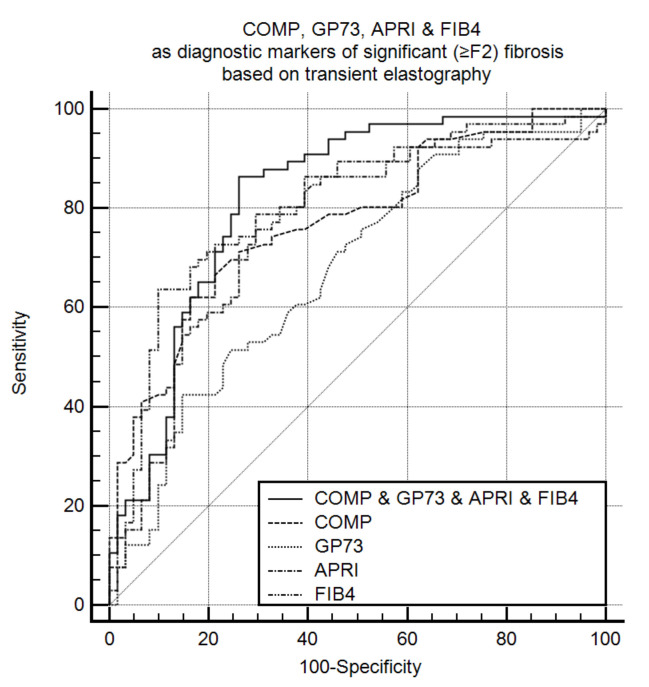
COMP, GP73, APRI, and FIB-4 as diagnostic markers of significant fibrosis (≥F2) according to transient elastography (*n* = 127). The combination of COMP, GP73, and FIB-4 had the highest diagnostic accuracy [AUC (95% CI): 0.841 0.766–0.900)] compared to each marker alone: GP73 [AUC (95% CI): 0.676 (0.587–0.756); *P* < 0.001], COMP [AUC (95% CI): 0.772 (0.690–0.842); *P* = 0.100], APRI [AUC (95% CI): 0.759 (0.675–0.830); *P* = 0.052], and FIB-4 [AUC (95% CI): 0.802 (0.722–0.868); *P* = 0.210]. COMP: Cartilage oligomeric matrix protein. GP73: Golgi protein 73. APRI: Aspartate aminotransferase to platelets index. FIB-4: Fibrosis-4. AUC: Area under the curve.

**Figure 5 cancers-13-03510-f005:**
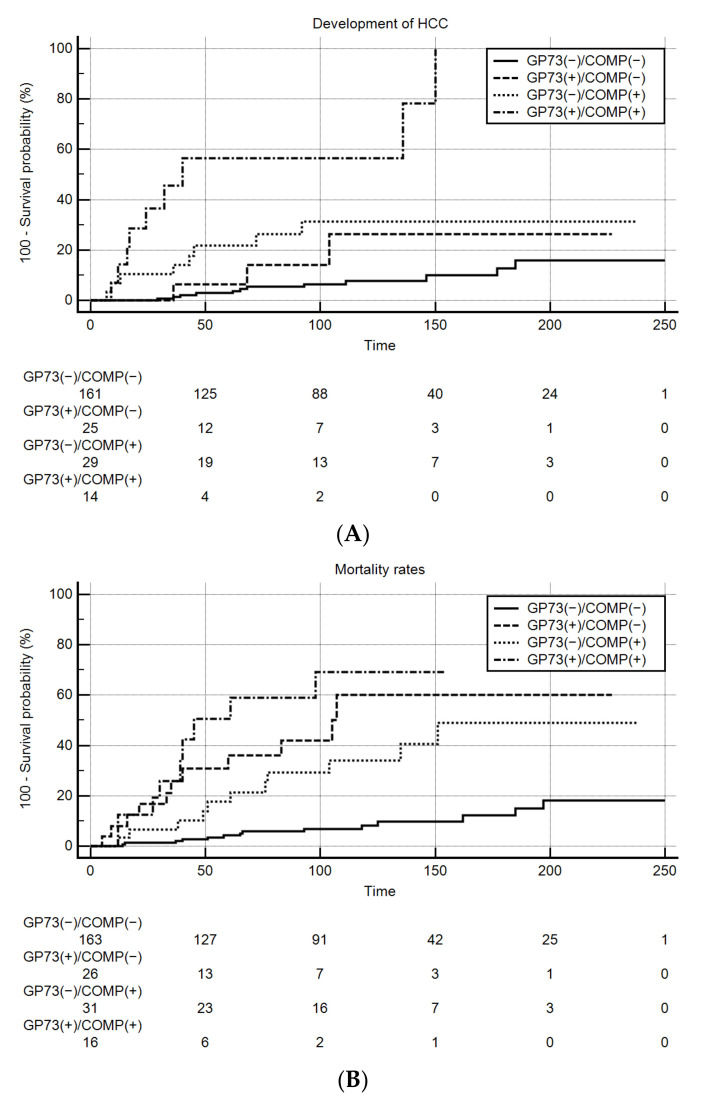
Kaplan-Meier analysis for COMP and GP73 as predictive markers of HCC development and survival. GP73(+)/COMP(+) patients characterized by: (**A**) Higher rates of HCC development during follow-up (*P* < 0.001; analysis was performed in 229 patients with follow-up more than 6 months by excluding patients with HCC at baseline), (**B**) increased liver-related mortality (*P* < 0.001).

**Table 1 cancers-13-03510-t001:** Baseline demographic, biochemical, histological, and clinical characteristics of patients.

Baseline Characteristics	Total Cohort (*n* = 288)
Age, median (IQR), years	53 (25)
Sex, male/female, *n* (%)	161/127 (55.9%/44.1%)
CHB/CHC/Alcoholic/PBC/AIH, *n*	129/120/12/21/6
GP73, median (IQR), units, (ULN: 20 units)	10.4 (9)
GP73, pos/neg, *n* (%)	54/234 (18.8%/81.3%)
COMP, median (IQR), units, (ULN: 15 U/L)	10.3 (6)
COMP, pos/neg, *n* (%)	61/227 (21.2%/78.8%)
INR, median (IQR), (normal: 0.85–1.15)	1.04 (0.17)
Platelets, median (IQR), ×10^3^/μL (normal: 140–440)	190 (100)
AST, median (IQR), (ULN: 40 U/L)	35 (40)
ALT, median (IQR), (ULN: 40 U/L)	39 (44)
γ-GT, median (IQR), (ULN: 37 U/L)	32 (52)
ALP, median (IQR), (ULN: 104 U/L)	81 (60)
Bilirubin, median (IQR), (ULN: 1.1 mg/dL)	0.8 (0.7)
Albumin, median (IQR), (normal: 3.5–5.2 g/dL)	4.4 (0.8)
AFP, median (IQR), (ULN: 10 ng/mL)	3.4 (4.5)
APRI	0.5 (0.7)
FIB-4	1.5 (2)
METAVIR F1/F2/F3/F4, *n* (%) *	63/44/20/50 (35.6%/24.9%/11.3%/28.2%)
Baseline cirrhosis, *n* (%)	116/288 (40.3%)
Baseline decompensation of cirrhosis, *n* (%)	33/116 (28.4%)
Baseline HCC, *n* (%)	12/288 (4.2%)
Follow-up ≥ 6 months, yes/no, *n* (%)	236/52 (81.9%/18.1%)
Duration of follow-up, median (IQR), months	98 (92)

* According to liver biopsy (available in 177 patients). IQR: Interquartile range. CHB: Chronic hepatitis B. CHC: Chronic hepatitis C. PBC: Primary biliary cholangitis. AIH: Autoimmune hepatitis. GP73: Golgi protein 73. ULN: Upper limit of normal. COMP: Cartilage oligomeric matrix protein. INR: International normalized ratio. AST: Aspartate aminotransferase. ALT: Alanine aminotransferase. γ-GT: γ-glutamyl transpeptidase. ALP: Alkaline phosphatase. AFP: Alpha-fetoprotein. FIB-4: Fibrosis-4 score. APRI: aspartate aminotransferase/platelets ratio index. HCC: Hepatocellular carcinoma.

**Table 2 cancers-13-03510-t002:** Baseline demographic, clinical, and laboratory characteristics of patients according to COMP and GP73 positivity.

Baseline Characteristics	GP73 Pos./COMP Pos. (*n* = 24)	GP73 Pos./COMP Neg. (*n* = 30)	GP73 Neg./COMP Pos. (*n* = 37)	GP73 Neg./COMP Neg. (*n* = 197)	*P*-Value
Age, median (IQR), years	65 (13)	55 (26)	58 (21)	51 (23)	<0.001
INR, median (IQR), (normal: 0.85–1.15)	1.25 (0.28)	1.21 (0.44)	1.09 (0.22)	1.01 (0.11)	<0.001
Platelets, median (IQR), ×10^3^/μL (normal: 140–440)	87 (67)	154 (96)	185 (110)	199 (79)	<0.001
AST, median (IQR), (ULN: 40 U/L)	91 (90)	42 (51)	38 (39)	28 (29)	<0.001
ALT, median (IQR), (ULN: 40 U/L)	72 (88)	43 (60)	35 (32)	37 (41)	<0.001
γ-GT, median (IQR), (ULN: 37 U/L)	61 (86)	47 (112)	36 (44)	27 (45)	<0.001
ALP, median (IQR), (ULN: 104 U/L)	144 (111)	90 (62)	93 (63)	72 (57)	<0.001
Bilirubin, median (IQR), (ULN: 1.1 mg/dL)	1.9 (3.2)	1 (1.4)	0.8 (0.8)	0.7 (0.6)	<0.001
Albumin, median (IQR), (normal: 3.5–5.2 g/dL)	3.5 (1.1)	3.8 (0.9)	4.3 (1)	4.5 (0.6)	<0.001
AFP, median (IQR), (ULN: 10 ng/mL)	9.9 (20.4)	4.3 (10.8)	4.8 (6)	3 (3)	<0.001
APRI	2.7 (3.8)	0.8 (1.2)	0.6 (0.6)	0.4 (0.5)	<0.001
FIB-4	8.3 (6.3)	3.4 (3.8)	2.1 (2.1)	1.3 (1.2)	<0.001
METAVIR F2-F3-F4/F1, *n* (%) *	8 (100%)/0 (0%)	14 (93.3%)/1 (6.7%)	16 (94.1%)/1 (5.9%)	76 (55%)/61 (44%)	<0.001
Baseline cirrhosis, *n* (%)	23 (95.8%)	22 (73.3%)	25 (67.6%)	46 (23.4%)	<0.001
Baseline decompensation of cirrhosis ^#^, *n* (%)	9 (39.1%)	11 (50%)	5 (20%)	8 (17.4%)	0.019
Baseline HCC, *n* (%)	2 (8.3%)	3 (10%)	4 (10.8%)	3 (1.5%)	0.011

* According to liver biopsy available in 177 patients. ^#^ Among 116 patients who were cirrhotic at baseline. GP73: Golgi protein 73. ULN: Upper limit of normal. COMP: Cartilage oligomeric matrix protein. IQR: Interquartile range. INR: International normalized ratio. AST: Aspartate aminotransferase. ALT: Alanine aminotransferase. γ-GT: γ-glutamyl transpeptidase. ALP: Alkaline phosphatase. AFP: Alpha-fetoprotein. APRI: aspartate aminotransferase/platelets ratio index. FIB-4: Fibrosis-4 score. HCC: Hepatocellular carcinoma.

**Table 3 cancers-13-03510-t003:** Baseline factors associated with HCC development during follow-up.

Baseline Factors	Univariate Analysis			Multivariate Analysis		
	HR	95% CI	*P*-Value	HR	95% CI	*P*-Value
COMP + GP73	39.81	12.60–125.7	<0.001	13.55	3.616–50.78	<0.001
Age	1.106	1.066–1.148	<0.001	1.079	1.038–1.122	<0.001
Male sex	3.168	1.368–7.337	0.007	3.134	1.338–7.340	0.009
Platelets	0.033	0.008–0.130	<0.001	19.78	0.749–522.6	ns
AFP	0.999	0.990–1.008	0.814	na	na	na
FIB-4	1.286	1.206–1.372	<0.001	1.302	1.089–1.557	0.004
Viral vs. non-viral liver disease	1.595	0.483–5.265	0.443	na	na	na

HCC: Hepatocellular carcinoma. COMP: Cartilage oligomeric matrix protein. GP73: Golgi protein 73. AFP: Alpha-fetoprotein. HR: Hazard ratio. CI: Confidence interval. na: not applicable. ns: not statistically significant.

**Table 4 cancers-13-03510-t004:** Baseline factors associated with liver-related mortality.

Baseline Factors	Univariate Analysis			Multivariate Analysis		
	HR	95% CI	*P*-Value	HR	95% CI	*P*-Value
COMP + GP73	62.48	22.01–177.3	<0.001	23.11	6.972–76.58	<0.001
Age	1.081	1.051–1.113	0.001	1.044	1.014–1.076	0.004
Male sex	1.539	0.835–2.837	0.167	na	na	na
Platelets	0.035	0.011–0.108	<0.001	0.817	0.078–8.559	0.866
FIB-4	1.228	1.165–1.294	<0.001	1.073	0.944–1.220	0.134
Viral vs. non-viral liver disease	0.434	0.223–0.844	0.014	1.073	0.944–1.220	0.280

COMP: Cartilage oligomeric matrix protein. GP73: Golgi protein 73. AFP: Alpha-fetoprotein. HR: Hazard ratio. CI: Confidence Interval. na: non-applicable.

## Data Availability

The data presented in this study are available on request from the corresponding authors.

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
