# Peer review of "Serum Cartilage Oligomeric Matrix Protein and Golgi Protein-73: New Diagnostic and Predictive Tools for Liver Fibrosis and Hepatocellular Cancer?"

_cancers, 2021, doi:10.3390/cancers13143510_

Round 1

Reviewer 1 Report

In this study, Gatselis NK et al., have evaluated the diagnostic and prognostic values of serum COMP and GP73 levels in chronic liver disease patients. They showed that their combination together with APRI showed the best performance to diagnose liver fibrosis and that their combination could predict the future HCC occurrence and high mortality.

Development of useful biomarkers for diagnosing advanced liver fibrosis and predicting future liver-related lethal events is unmet need. Thus, this paper could be potentially interesting. However, the analysis presented in this paper is still preliminary and need improvement in order to claim the usefulness of their combined marker. Here is the specific comments.

1) In the comparison of diagnostic ability of liver fibrosis, all the possible combinations and their comparisons should be shown (i.e. COMP + APRI, GP73 + APRI). In addition, FIB-4 and other typical fibrosis markers such as hyaluronic acid or type IV collagen should be also included for the evaluation. Formula for each combination also need to be presented.

2)In general, advanced fibrosis consider F>3. What about the ability of COMP and GP73 for the diagnosis of F>3 fibrosis?

3)In the KM analysis of HCC occurrence, what if you include single GP73(+) group and single COMP(+) group for the comparison? Are there any additive effect of their combination?

4)Please perform multivariate analysis using Logistic regression for diagnosis of liver fibrosis and COX proportional hazard for HCC occurrence and mortality. These analysis can find the confounding factors and thus really assess the diagnostic and prognostic value of combination of COMP and GP73.  

5) I believe that log-rank test is not appropriate for the comparison of ROC. It should be done by De Long test or other appropriate ones.

Author Response

Comments and Suggestions for Authors

Here is the specific comments.

1. “In the comparison of diagnostic ability of liver fibrosis, all the possible combinations and their comparisons should be shown (i.e. COMP + APRI, GP73 + APRI). In addition, FIB-4 and other typical fibrosis markers such as hyaluronic acid or type IV collagen should be also included for the evaluation. Formula for each combination also need to be presented”.

Reply: I would like to thank the reviewer for this valid comment. Following your suggestion, we included in our analysis Fibrosis-4 score (FIB-4) and we modified accordingly the Section 3.2 and Figures 2-4. In addition, all possible various combinations along with their comparisons and formulas are now presented in a new Supplementary Table 1.

2. “In general, advanced fibrosis consider F>3. What about the ability of COMP and GP73 for the diagnosis of F>3 fibrosis?”

Reply: According to your recommendation we now performed a sub-analysis which showed that the quadrable combination of COMP, GP73, APRI and FIB-4 retained the highest diagnostic accuracy to detect severe (≥F3) fibrosis (Section 3.2, 2nd paragraph, p. 7 and Supplementary Table 1), while the triple combination of COMP, GP73 and FIB4 had the highest ability to detect ≥F3 fibrosis according to LSM determined by transient elastography (Section 3.2, 3rd paragraph, p. 8 and Supplementary Table 1).

3. “In the KM analysis of HCC occurrence, what if you include single GP73(+) group and single COMP(+) group for the comparison? Are there any additive effect of their combination?”

Reply: We have followed your suggestion and we performed two separate Kaplan-Meier analyses, where we compared GP73(+)/COMP(+) vs. GP73(+)/COMP(-) and GP73(+)/COMP(+) vs. GP73(-)/COMP(+) patients. In both analyses, the dual positivity has the highest ability to predict the development of HCC during follow-up than single positivity for either GP73 or COMP alone (Section 3.3, 2nd paragraph, p.10).

4. “Please perform multivariate analysis using Logistic regression for diagnosis of liver fibrosis and COX proportional hazard for HCC occurrence and mortality. These analysis can find the confounding factors and thus really assess the diagnostic and prognostic value of combination of COMP and GP73”.

Reply: A multivariate logistic regression analysis confirmed the diagnostic ability of COMP plus GP73 combination for diagnosing cirrhosis after adjustment for age, male sex, AST/ALT levels, platelets, γGT and categorization of liver disease as viral or non-viral (Section 3.1, 5th paragraph, p. 6). Besides, a multivariate Cox regression analysis showed that the combination of COMP and GP73 was the strongest negative predictive factor for the development of HCC and liver-related mortality, after adjustment for age, male sex, platelet levels, AFP and categorization of liver disease as viral or non-viral (Section 3.3, 2nd and 5th Paragraph, p.10).

 5. “I believe that log-rank test is not appropriate for the comparison of ROC. It should be done by De Long test or other appropriate ones”.

Reply: Thank you very much indeed for this important comment. Of course, the DeLong test is the appropriate test for the comparison of ROC curves and this has been done but mistakenly we wrote the log-rank test. The correction is now clear (Section of statistical analysis). Thank you again for this very significant comment that precluded our mistake.

Reviewer 2 Report

Gatselis et al. evaluated the usefulness of cartilage oligomeric matrix protein and Golgi protein-73 as diagnostic and predictive tools for liver fibrosis and hepatocellular carcinoma.

The paper is well written and organized, and the rationale of the study is clear. I assume the authors reported the paper following the STARD statement. Here are some of the comments.

APRI is now regarded as a fibrosis marker in daily clinical practice; however, readers will wonder about the performance of the FIB-4 index. Please consider adding FIB-4 in the analysis. The cut-off was used from their previous papers. The authors should perform the analysis with continuous valuable rather than categorical valuable or different cut-offs from other published papers.

When did the authors recruit patients? Since it may affect antiviral treatment such as IFN, DAA, NA and may eventually affect HCC development and decompensation.

What is the mean follow-up period?

Please provide the IRB approval number.

Risk factors for HCC development should be analyzed by Cox proportional hazard regression model.

Were there any patients who underwent liver transplantation? Please clarify.

Author Response

The paper is well written and organized, and the rationale of the study is clear. I assume the authors reported the paper following the STARD statement. Here are some of the comments.

1. “APRI is now regarded as a fibrosis marker in daily clinical practice; however, readers will wonder about the performance of the FIB-4 index. Please consider adding FIB-4 in the analysis”.

Reply: I can understand the concerns raised by the reviewer. According to your suggestion, we have now included in our analysis the Fibrosis-4 score (FIB-4). Please see our responses to Reviewer’s 1 comments above.

 2. “The cut-off was used from their previous papers. The authors should perform the analysis with continuous valuable rather than categorical valuable or different cut-offs from other published papers”.

Reply: Following your suggestion, we performed an additional analysis using COMP and GP73 levels as continuous variables (the results are presented in Section 3.1, 1st and 2nd paragraph). However, we also keep the cut-off of our previous publications (GP73 >20 units and COMP >15 U/L), as they have been precisely validated. Regarding GP73, it has been reported in detail in our previous publication that the cut-off of 20 units was established by applying the test to a very large population of 175 healthy and 1316 disease controls including patients with anti- phospholipid syndrome (n=25), rheumatoid arthritis (n=55), scleroderma (n=25), inflammatory bowel disease (n=10), breast cancer (n=46), colorectal cancer (n=50), prostate cancer (n=45), non-cirrhotic chronic hepatitis C (n=14), non-cirrhotic chronic hepatitis B (n=25), human immunodeficiency virus infection (n=8), syphilis (n=40), cirrhosis (n=719) and hepatocellular carcinoma (n=254) [Gatselis et al, World J Gastroenterol 2020]. Similarly, for COMP the cut-off level of 15 U/L was determined after thorough examinations in normal as well as disease controls [please see Norman et al, World J Hepatol 2015; Zachou et al, Eur J Intern Med 2017].

3. “When did the authors recruit patients? Since it may affect antiviral treatment such as IFN, DAA, NA and may eventually affect HCC development and decompensation”.

Reply: Thank you again for your comment. Our patients recruited between 2000 and 2013 (please see p.2, last par.). However, the very long follow-up period up to the first trimester of 2021, with a median (IQR) duration longer than 8 years seems to ameliorate the impact of the different treatment strategies for patients with viral hepatitis. This comment has now been clearly added in the discussion section before the conclusion (please see p. 12 and 13).

4. “What is the mean follow-up period?”

Reply: We have followed your recommendation and the required information has now been added in the Material and Methods, 2nd Paragraph.

 5. “Please provide the IRB approval number”.

Reply: The respective number of approval by the ethical committee of the General University Hospital of Larissa had already been included in the previous version of the manuscript (p. 10) and is also included now in the revised version (p.13; Institutional Review Board Statement).

6. “Risk factors for HCC development should be analyzed by Cox proportional hazard regression model.

Reply: We would like to thank the reviewer for his/her valid comment. Accordingly, a multivariate COX-regression analysis was performed and revealed that the combination of COMP and GP73 was the strongest negative predictive factor for the development of HCC, after adjustment for age, male sex, platelet levels, AFP and categorization of liver disease as viral or non-viral (please see Section 3.3, 2nd Paragraph).

7. “Were there any patients who underwent liver transplantation? Please clarify”.

Reply: None of the patients enrolled in this study was underwent liver transplantation during the follow-up period of the study (please see p. 3, last sentence).

Reviewer 3 Report

The manuscript “Serum cartilage oligomeric matrix protein and Golgi protein-73: New diagnostic and predictive tools for liver fibrosis and hepatocellular cancer?” by Gatselis et al. aimed at finding combination markers for the progression of liver diseases to Fibrosis/ cirrhosis and hepatocellular carcinoma. They evaluated the performance of two biomarkers, cartilage oligomeric matrix protein (COMP) and Golgi protein-73 (GP73), individually associated with the progression of chronic liver diseases, to detect fibrosis/cirrhosis and predict the development of HCC. In a cohort of 288 patients with chronic liver disease, the authors found that the combination GP73 and COMP had a high discriminative ability to detect severe fibrosis/cirrhosis and efficiently predict HCC.

Major comments:

  1. Did the authors look at the AFP-negative patients to validate that the combined positivity of COM and GP73 is helpful for early detection of fibrosis/cirrhosis and the prediction of HCC in this population?
  2. Would the etiology of liver disease affect the combined positivity of COM/GP73 as a biomarker?
  3. Can the authors discuss the usefulness of the others biomarkers combinations discussed in the literature?
  4. How the triple combined serum positivity for COM/GP73/AFP affects the sensitivity for early detection of HCC?
  5. Can the authors discuss COM specificity (liver vs. other tissues) in the liver-specific diagnosis?
  6. The authors discussed COM/GP73 positivity and the sensitivity of detection/prediction of cirrhosis/HCC. Would an increase in COM/GRP levels or their levels ratio increase the prediction of HCC?

Minor Comments

  1. Some typos in the text.
  2. Table 2, the first row, needs to be made easier to read.

Author Response

Major comments:

1. “Did the authors look at the AFP-negative patients to validate that the combined positivity of COM and GP73 is helpful for early detection of fibrosis/cirrhosis and the prediction of HCC in this population?”

Reply: According to your suggestion, we compared the diagnostic accuracy of COMP and GP73 combination for the detection of cirrhosis and severe fibrosis (≥F3) based on liver biopsy or transient elastography between patients with increased AFP (10 ng/mL) and normal AFP (≤10 ng/mL). The results showed that AFP levels did not affect the accuracy to diagnose fibrosis/cirrhosis (please see p.9; 1st paragraph). Similarly, the value of COMP and GP73 as a predictor of HCC development remained significant in patients with normal AFP (please see Section 3.3, 2nd paragraph, last sentence).

2. “Would the etiology of liver disease affect the combined positivity of COM/GP73 as a biomarker?”

Reply: We would like to thank the reviewer for this important note. To investigate any impact of the etiology of liver disease, we grouped our patients as viral and non-viral. Multivariate logistic regression analysis did not render this categorization as a significant factor for the presence of cirrhosis (please see Section 3.1, 5th paragraph on p. 6). Similarly, multivariate COX-regression analysis showed that the etiology of liver disease (viral vs. non-viral) did not impact the development of HCC (please see Section 3.3, 2nd Paragraph on p. 10) or survival (please see Section 3.3, 5th Paragraph on p. 10).

3. “Can the authors discuss the usefulness of the others biomarkers combinations discussed in the literature?”

Reply: Thank you again for your comment. Following your recommendation, we have now added a brief comment on the usefulness of other biomarkers combinations with the appropriate new references (please see the revised Discussion section, 5th paragraph on p. 12).

4. How the triple combined serum positivity for COM/GP73/AFP affects the sensitivity for early detection of HCC?”

Reply: This is a very important comment raised by the reviewer. Thank you. In order to satisfy the reviewer, we performed two separate analyses in an attempt to assess the effectiveness of the triple combination of COMP/GP73/AFP serum positivity: (i) At baseline, 12 patients were diagnosed with HCC. The diagnostic performance of AFP to diagnose HCC showed 67% sensitivity, 89% specificity, 21% positive predictive value, 98% negative predictive value. The triple combined serum positivity for COMP, GP73 and AFP showed 17% sensitivity, 98% specificity, 25% positive predictive value and 96% negative predictive value (please see Section 3.1, 6th paragraph on p.6).  (ii) Among 229 patients without baseline HCC who were followed, AFP positivity was characterized by 41% sensitivity, 91% specificity, 42% positive predictive value and 90% negative predictive value for the development of HCC during follow-up. The triple combined serum positivity for COMP, GP73 and AFP demonstrated 13% sensitivity, 99% specificity, 67% positive predictive value and 87% negative predictive value for HCC development (please see Section 3.3, 3rd Paragraph on p.10).

5. “Can the authors discuss COM specificity (liver vs. other tissues) in the liver-specific diagnosis?”

Reply: Again, this is a very valid comment. Thank you. According to your recommendation we have now discussed in more detail about issues that may raise regarding COMP specificity in liver specific diagnosis, as it was considered that it mainly originates from cartilage tissue (please see the revised Discussion section, 4th paragraph on p. 12). In addition, as in our previous studies in the field and in an attempt to minimize the likelihood of false positive results in our current study, we would like to make clear that we have excluded patients with concurrent history of autoimmune rheumatic diseases or osteoarthritis at baseline and during follow-up. This statement which had been forgotten in the previous version of the manuscript has now been added and we would like to thank you for reminding us this very important issue (please see Material and Methods, 1st paragraph on p. 2).

6. “The authors discussed COM/GP73 positivity and the sensitivity of detection/prediction of cirrhosis/HCC. Would an increase in COM/GRP levels or their levels ratio increase the prediction of HCC?”

Reply: Thank you again for the valid comment. To investigate if an increase of COMP or GP73 levels increases further the likelihood for HCC development we stratified each biomarker into 3 categories: (i) GP73 or COMP ≤ mean+SD, (ii) mean+SD < GP73 or COMP ≤ mean+2SD, (iii) GP73 or COMP > mean+2SD.  By using GP73 ≤ mean+SD as reference group, we found that increase of GP73 levels raises the likelihood of HCC future development: HR=7.703, 95%CI: 2.849-20.82 for “GP73 > mean+SD” group (P<0.001) and HR=10.8, 95%CI:3.699-31.543 for “GP73 > mean+2SD” group (P<0.001). Regarding COMP, further increase of likelihood for HCC development was noticed only for “COMP > mean+SD” group with HR=4.686, 95%CI: 2.012-10.91 (P<0.001) (please see Section 3.3, 4th paragraph on p. 10). In contrast, no significance was found about GP73/COMP ratio and the development of HCC (HR=1.075, 95%CI: 0.942-1.228, P=0.283)

Minor Comments

1. “Some typos in the text”.

Reply: The revised text has now been checked thoroughly and corrected

 2. “Table 2, the first row, needs to be made easier to read”.

Reply: We tried to change the respective row. Hope that it is much easier and clear now.

Round 2

Reviewer 1 Report

I don't think that authors appropriately addressed my concerns. The most important point for the biomarker study is whether identified markers significantly outperform current practical ones or not. In this point, FIB-4 is the most simple and reliable marker to diagnose advanced fibrosis and even predict future HCC development. So it is mandatory to show the performance of the new markers compared to it in every way. 

In this manuscript, authors would like to claim the usefulness of the combination of GP73 and COMP. In that case, authors must show the superiority for this combination to FIB-4 or at least the superiority of the trio (GP73, COMP, and FIB-4) to FIB-4. Since FIB-4 and APRI use the same parameters, combination of those is not acceptable. Trio (GP73, COMP, and APRI) versus FIB-4 may be also acceptable.

Regarding multivariate COX regression analysis for the predictors of liver-related mortality, they must include FIB-4 in variables, which can truly evaluate the predictive performance of your combo. You also need to provide table including univariate and multivariate COX regression analysis of all the important clinical variables.

Author Response

REVIEWER 1

Comments and Suggestions for Authors

Comment 1: “I don't think that authors appropriately addressed my concerns. The most important point for the biomarker study is whether identified markers significantly outperform current practical ones or not. In this point, FIB-4 is the most simple and reliable marker to diagnose advanced fibrosis and even predict future HCC development. So, it is mandatory to show the performance of the new markers compared to it in every way. In this manuscript, authors would like to claim the usefulness of the combination of GP73 and COMP. In that case, authors must show the superiority for this combination to FIB-4 or at least the superiority of the trio (GP73, COMP, and FIB-4) to FIB-4. Since FIB-4 and APRI use the same parameters, combination of those is not acceptable. Trio (GP73, COMP, and APRI) versus FIB-4 may be also acceptable”.

Reply: Thank you for your comment. We agree with you that novel biomarkers such as COMP and GP73 should be compared to currently used biomarkers such as APRI and FIB-4. For this reason and following your suggestion during the first review of our manuscript, we constructed the Supplementary Table 1 to show the diagnostic performance by using the AUC (95%CI) of all biomarkers as single markers or their dual, triple and quadrable combinations.

To make it solid clear, in Supplementary Table 1, the P-values were calculated by comparing the combination with the highest AUC as reference (actually the quadrable combination), as it is not feasible to present the P-values off all possible comparisons.

Moreover, following your new instructions after the second review of our paper, we added in the main text of the paper the diagnostic performance [AUC (95%CI)] of the triple combination of COMP, GP73 and FIB-4 and compared this to each marker alone (COMP, GP73, COMP and FIB-4). Please see the 1st and 2nd paragraphs of the revised Section 3.2.

Although APRI and FIB-4 formulas share AST and platelets, FIB-4 also includes the age and ALT values. This difference allows the use of dual combination of APRI and FIB-4 to predict cirrhosis and also improves their diagnostic accuracy compared to each biomarker alone (Papadopoulos et al, Ann Gastroenterol 2019; Oliveira et al, Ann Hepatol 2016; Tural, Clin Gastroenterol Hepatol 2009). This is in line with our findings where the quadrable combination achieved a little bit better performance compared to the triple combination (COMP+GP73+FIB-4).

Comment 2: “Regarding multivariate COX regression analysis for the predictors of liver-related mortality, they must include FIB-4 in variables, which can truly evaluate the predictive performance of your combo. You also need to provide table including univariate and multivariate COX regression analysis of all the important clinical variables”.

Reply: We really appreciate your comment. We provide now in the re-revised version of our paper the new Table 4 in Section 3.3 with the results of the univariate and multivariate analysis. Following your thoughtful suggestion for the inclusion of FIB-4 in variables, we found again that the combination of COMP with GP73 was the most potent negative predictive factor of liver related mortality [HR 23.11, 95%CI6.972-76.58; P<0.001], while FIB4 did not reach a statistical significance in the multivariate analysis. The 5th paragraph of Section 3.3 was modified accordingly with the new results.

In order to satisfy further the reviewer, a similar addition was also done regarding the COMP+GP73 combination as a predictive factor for HCC development (new Table 3). As you can see the combination remained the strongest negative predictive factor for HCC development even after the inclusion of FIB-4 in the analysis.

Reviewer 2 Report

The authors fulfilled each of the compulsory revisions and modified the manuscript as requested. However, I have the following request that will further improve the paper.

Please include tale 3 with HR, CI, the p-value of cox regression analysis.

Author Response

REVIEWER 2

Comments and Suggestions for Authors

The authors fulfilled each of the compulsory revisions and modified the manuscript as requested. However, I have the following request that will further improve the paper.

Comment: “Please include table 3 with HR, CI, the p-value of cox regression analysis”.

Reply: Thank you for your comment. Please refer to our responses to reviewer 1 above.   

Round 3

Reviewer 1 Report

Revised manuscript is of greatly improved quality. I believe it is worthy of being published. I don't have any further comment.